# DeepTAGE: Deep Temporal-Aligned Gradient Enhancement for Optimizing Spiking Neural Networks

**Wei Liu[1], Li Yang[1], Mingxuan Zhao[1], Shuxun Wang[1], Jin Gao[1], Wenjuan Li[1]**
Bing Li[1], Weiming Hu[1,2,3*]

[1]Beijing Key Laboratory of Super-Intelligent Security of Multi-Modal Information, MAIS, CASIA
[2]School of Artificial Intelligence, University of Chinese Academy of Sciences
[3]School of Information Science and Technology, ShanghaiTech University
`zhaomingxuan@bupt.edu.cn`
`{liuwei,wangshuxun2022}@ia.ac.cn`
`{li.yang,jin.gao,wenjuan.li,bli,wmhu}@nlpr.ia.ac.cn`

## Abstract

Spiking Neural Networks (SNNs), with their biologically inspired spatio-temporal dynamics and spike-driven processing, are emerging as a promising low-power alternative to traditional Artificial Neural Networks (ANNs). However, the complex neuronal dynamics and non-differentiable spike communication mechanisms in SNNs present substantial challenges for efficient training. By analyzing the membrane potentials in spiking neurons, we found that their distributions can increasingly deviate from the firing threshold as time progresses, which tends to cause diminished backpropagation gradients and unbalanced optimization. To address these challenges, we propose Deep Temporal-Aligned Gradient Enhancement (DeepTAGE), a novel approach that improves optimization gradients in SNNs from both internal surrogate gradient functions and external supervision methods. Our DeepTAGE dynamically adjusts surrogate gradients in accordance with the membrane potential distribution across different time steps, enhancing their respective gradients in a temporal-aligned manner that promotes balanced training. Moreover, to mitigate issues of gradient vanishing or deviating during backpropagation, DeepTAGE incorporates deep supervision at both spatial (network stages) and temporal (time steps) levels to ensure more effective and robust network optimization. Importantly, our method can be seamlessly integrated into existing SNN architectures without imposing additional inference costs or requiring extra control modules. We validate the efficacy of DeepTAGE through extensive experiments on static benchmarks (CIFAR10, CIFAR100, and ImageNet-1k) and a neuromorphic dataset (DVS-CIFAR10), demonstrating significant performance improvements.

## 1 Introduction

Spiking Neural Networks (SNNs) are a biologically inspired computing paradigm designed with dynamic spatio-temporal connections and binary spike-driven communication mechanisms (Roy et al., 2019; Schuman et al., 2022; Li et al., 2023). Unlike traditional Artificial Neural Networks (ANNs), which rely on continuous activations, SNNs operate with discrete spikes, offering the potential for more efficient and energy-saving computations, particularly in neuromorphic hardware implementations. The event-based nature of SNNs, where computations are triggered only upon the receipt of a spike, provides inherent sparsity and lower power consumption, making them attractive for real-time applications in energy-constrained environments (Pei et al., 2019; Davies et al., 2018; Merolla et al., 2014; Frenkel et al., 2023).

---

*Corresponding author: wmhu@nlpr.ia.ac.cn

However, despite these advantages, the complex dynamics of SNNs pose significant challenges for effective training. The simultaneous propagation of backpropagation gradients through both spatial and temporal dimensions is complicated by the non-differentiable nature of the spike activation function. As a result, training SNNs remains considerably more difficult than training ANNs. To mitigate this issue, previous works have employed surrogate gradient methods to approximate gradients during backpropagation by replacing the non-differentiable spike activation function with smoother alternatives. This approach has achieved strong results in shallow networks but faces challenges when scaling to deeper networks due to the vanishing gradient problem. While various methods, including residual connections (He et al., 2016b;a; Fang et al., 2021a; Zheng et al., 2021; Hu et al., 2024), normalization techniques (Kim & Panda, 2021), and attention mechanisms (Duan et al., 2022; Yao et al., 2023c;a;b), have been adopted for improved SNN learning, they do not adequately address the challenges of maintaining effective gradient propagation across both spatial and temporal dimensions in SNNs.

In this work, we focus on optimizing SNNs by enhancing their backpropagation gradients across spatial and temporal dimensions. Our analysis reveals that as the time step increases, the distribution of membrane potentials in SNNs tends to shift away from the firing threshold, leading to diminished gradient magnitudes and unbalanced training processes. This shift can further exacerbate the vanishing gradient problem, particularly during the backpropagation of gradients through multiple network layers and time steps.

To address these challenges, we propose **Deep Temporal-Aligned Gradient Enhancement** (Deep-TAGE), a novel method that enhances the optimization gradients of SNNs from both the internal surrogate gradient functions and external supervision approaches. DeepTAGE introduces a temporal alignment mechanism that dynamically adjusts the surrogate gradient functions based on the membrane potential distribution at each time step, ensuring that neurons receive more effective gradients when their membrane potentials diverge from the firing threshold. Additionally, we incorporate Spatio-Temporal Deep Supervision (STDS), which supplements deep supervision at multiple network stages and time steps of SNNs, further enhancing gradient flow and reducing optimization bottlenecks. Our gradient enhancement approach brings several key advantages: (1) improving spiking activity across network layers and time steps, leading to enhanced representation capabilities; (2) accelerating the training convergence of SNNs and ultimately achieving better performance; (3) improving SNN optimization without the need to change the network architecture or add additional control modules.

We validate the effectiveness of DeepTAGE through extensive experiments on both static and neuromorphic datasets, including CIFAR10 (Krizhevsky et al., 2010), CIFAR100 (Krizhevsky et al., 2010), ImageNet-1k (Deng et al., 2009), and DVS-CIFAR10 (Li et al., 2017). Across these benchmarks, our method consistently outperforms existing state-of-the-art methods, confirming its efficacy in optimizing SNNs for various tasks. Our main contributions are threefold:

- Temporal-Aligned Gradient Enhancement. We propose a novel gradient enhancement technique that adapts the surrogate gradient function according to the distribution of membrane potentials at each time step. This ensures more balanced and effective backpropagation throughout the network.

- Spatio-Temporal Deep Supervision. We introduce an optimization paradigm that supplements deep supervision into multiple network stages and time steps of SNNs, mitigating gradient vanishing issues and enhancing the convergence of the network.

- We validate the effectiveness of our methods through extensive experiments and demonstrate significant improvements over existing SNN methods on multiple benchmarks.

## 2 RELATED WORK

SNN optimization has progressed through two major avenues: ANN-to-SNN conversion and direct training via surrogate gradients. The ANN-to-SNN conversion approach hinges on approximating spiking neuron firing rates through ReLU activations in ANNs. Early work by Rueckauer et al. (2017) explored methods to minimize conversion errors by adjusting activation functions and introducing techniques like SpikeNorm and threshold balancing to reduce accuracy loss during conversion (Rueckauer et al., 2016; Cao et al., 2015; Sengupta et al., 2019). Despite these improvements,

conversion methods tend to require large time steps, which increases latency (Bu et al., 2023; Jiang et al., 2023).

Direct SNN training, on the other hand, utilizes surrogate gradients to handle the non-differentiability of spike functions, as seen in the spatiotemporal backpropagation (STBP) method introduced by Wu et al. (2018). This approach has achieved strong results in shallow networks but faces challenges when scaling to deeper networks due to vanishing gradients. To address this, several normalization techniques have been proposed, such as temporal batch normalization (BNTT) (Kim & Panda, 2021) and threshold-dependent batch normalization (tdBN) (Zheng et al., 2021), which improve gradient flow and convergence in deep SNNs.

Additionally, residual connections, inspired by ResNet architectures, have been adapted for SNNs. Fang et al. (2021a) proposed the SEW-ResNet framework, which enables more efficient spike propagation and helps maintain performance as network depth increases. Recent gradient optimization techniques, like Temporal Efficient Training(TET) (Deng et al., 2022) and Spatial Learning Through Time (SLTT) (Meng et al., 2023), further improve performance by refining backpropagation through time, with SLTT demonstrating that focusing on spatial gradients can boost optimization efficiency.

Our work builds on these foundations by introducing Deep Temporal-Aligned Gradient Enhancement (DeepTAGE), which adapts surrogate gradients according to the membrane potential distribution at each time step. This novel approach enhances training efficiency without adding computational complexity, improving SNN performance across various benchmarks.

# 3 METHOD

## 3.1 MOTIVATION

SNNs are characterized by their unique method of processing information through the temporal dynamics of membrane potential updates and spike activations across multiple time steps. These dynamics are crucial for understanding and optimizing the computational capabilities of SNNs. In Figure 1 (a), we present histograms that depict the distribution of membrane potentials at each time step within an inner layer of an SNN. These distributions evolve over time: they are sharply peaked and closer to the threshold voltage $v_{th}$ at initial time steps, but become broader and drift away from $v_{th}$ at later steps.

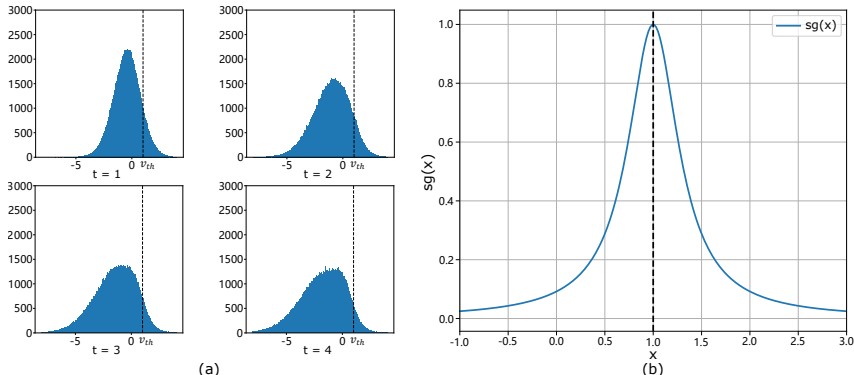

Figure 1: Visualization of membrane potential dynamics and surrogate gradient function in the spiking ResNet-19. (a) Histograms of membrane potentials at each time step from an inner layer of the ResNet-19, showing sharper distributions close to the threshold voltage ($v_{th}$) at earlier time steps and flatter distributions further from $v_{th}$ at later time steps. (b) Curve of the surrogate gradient function derived from the arctangent function, highlighting larger gradients near $v_{th}$ and smaller gradients as inputs deviate from $v_{th}$ ($v_{th} = 1$).

When training SNNs, due to the non-differentiable nature of the spike activation function for input membrane potentials, surrogate gradient functions are utilized to approximate the gradients necessary for effective backpropagation. Figure 1 (b) illustrates a commonly used surrogate gradient

function, which is derived from the derivative of an arctangent function (with $v_{th}$ set to 1):

$$\text{sg}(x) = \frac{1}{1 + \pi^2 (x - v_{th})^2}. \tag{1}$$

This function produces larger gradients when input values are proximal to $v_{th}$, with gradients decreasing as the values diverge from this threshold.

Considering the membrane potential distribution in Figure 1 (a), we can find that as time progresses, an increasing proportion of membrane potentials settle into regions where the surrogate gradients are negligible. This tends to result in an unbalanced optimization process for SNNs of multiple time steps and hinders the convergence of the network. Moreover, the vanishing gradient issue may be exacerbated as the network undergoes multiple layers of backpropagation, potentially impairing the network's spiking functionality and limiting its representational capabilities.

To tackle these challenges, we propose a novel approach called "Deep Temporal-Aligned Gradient Enhancement" to optimize SNNs. This approach aims to refine the gradient computation process and introduce deep supervision at both temporal and spatial levels. By balancing and enhancing the gradients over different time steps and network stages, our approach is expected to not only stabilize the learning process but also to amplify the representational power of SNNs by improving spike generation capability.

### 3.2 TEMPORAL-ALIGNED GRADIENT ENHANCEMENT

Based on the above analysis, it is natural to expect that the surrogate gradient functions should be varied at different time steps for better optimization of SNNs. In line with this, we propose a method termed Temporal-Aligned Gradient Enhancement (TAGE), which dynamically adapts the surrogate gradient function based on the distribution of input membrane potentials to balance gradient computation across various time steps.

To this end, we first model the degree of deviation of the input membrane potentials $x_t$ from the firing threshold $v_{th}$ at each time step $t$. We introduce a metric $\sigma_t$ which quantifies this deviation as the L2 norm of the difference between the potentials $x_t$ and threshold $v_{th}$:

$$\sigma_t = \|x_t - v_{th}\|_2. \tag{2}$$

A larger $\sigma_t$ reflects that more membrane potentials at time step $t$ are distributed away from the threshold $v_{th}$, often resulting in smaller gradients due to the properties of the surrogate gradient function described by equation 1.

As delineated in Section 3.1, the distribution of membrane potentials progressively shifts away from the threshold $v_{th}$ in subsequent time steps. This shift tends to cause unbalanced optimization and potentially exacerbate the vanishing gradient problem—a significant challenge when training SNNs over multiple time steps. To address this issue, we use $\sigma_t$ as a vital measure for comparing the distribution of membrane potentials across different time steps and adaptively modify the surrogate gradient function to ensure more uniform and effective training. The adaptation involves a normalization factor $\delta(\sigma_t)$, defined as follows:

$$\delta(\sigma_t) = w \cdot \frac{\sigma_t}{\sigma_1} + (1 - w), \tag{3}$$

where $w$ is a hyper-parameter that determines the impact of the distributional deviation ratio $\sigma_t / \sigma_1$ between times steps $t$ and 1, where $\delta(\sigma_t) = 1$ when $\sigma_t = \sigma_1$. This ratio ensures that the surrogate gradient function remains responsive to the changing distribution of inputs across time steps.

This normalization factor $\delta(\sigma_t)$ is integrated into the surrogate gradient function as follows:

$$\text{sg}_t(x_t) = \frac{1}{1 + \pi^2 \left( \frac{x_t - v_{th}}{\delta(\sigma_t)} \right)^2}. \tag{4}$$

This formula essentially scales the membrane potentials $x_t$ centered on $v_{th}$ when computing the surrogate gradients. The scaling does not change the size relationship between $x_t$ and $v_{th}$, and allows inputs with flatter distributions and farther away from $v_{th}$ to receive more effective gradients.

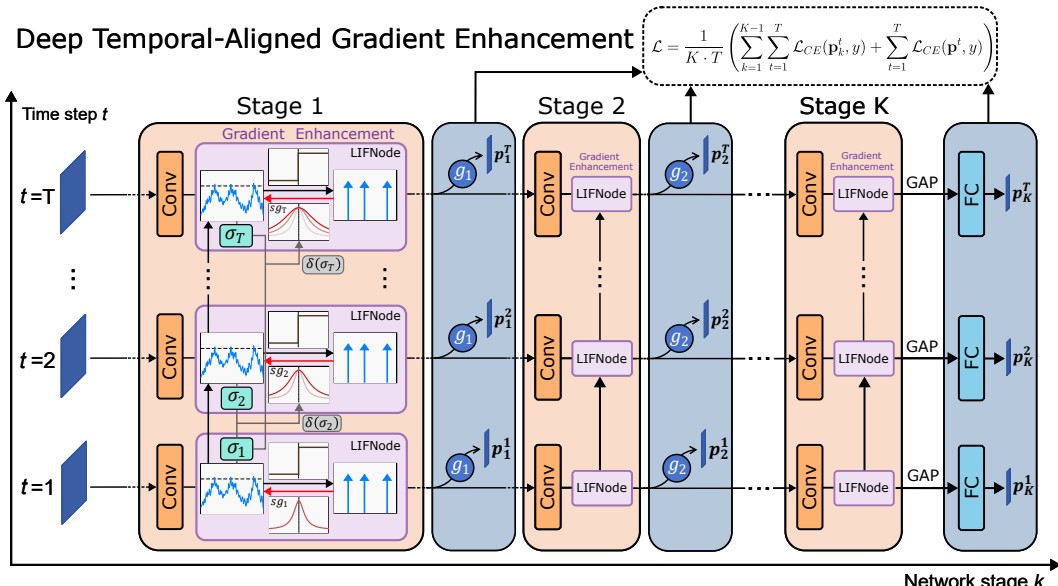

Figure 2: The overall framework of our Deep Temporal-Aligned Gradient Enhancement method. We propose to enhance gradient-based SNN optimization by adjusting the surrogate gradient functions of different time steps according to their membrane potential distributions. Our method also incorporates spatio-temporal deep supervision to further improve the gradient flow for SNN optimization.

## 3.3 SPATIO-TEMPORAL DEEP SUPERVISION

The proposed Temporal-Aligned Gradient Enhancement (TAGE) optimizes SNNs by adapting the surrogate gradient functions to the varying membrane potential distributions at different time steps. However, due to the inherent dynamic temporal dependence and the non-differentiable nature of activations within SNNs, gradients may vanish or deviate significantly as they back-propagate through multiple layers and time steps, which impedes the convergence and overall performance of the network. To tackle these challenges, we extend our approach to Deep Temporal-Aligned Gradient Enhancement (DeepTAGE), which introduce Spatio-Temporal Deep Supervision (STDS) into the network to enhance the gradient flow on the back-propagation path.

For an SNN with $K$ network stages, we denote its final classification output for each time step separately by $\{\mathbf{p}^t \mid t \in [1, T]\}$. To supplement deep supervision into the SNN, as illustrated in Figure 2, we introduce auxiliary classifier networks $\{g_k(\cdot) \mid k \in [1, K-1]\}$ at all network stages prior to the final $K$-th stage. Each classifier network consists of three convolution layers, followed by a global average pooling layer and a fully connected layer to produce classification predictions. Let $\{\mathbf{F}_k^t \mid t \in [1, T], k \in [1, K-1]\}$ denote the output feature maps of different time steps from previous $K-1$ network stages within the SNN. We feed these feature maps to the auxiliary classifier networks to produce extra classification predictions:

$$
\begin{aligned}
&\mathbf{p}_1^1 = g_1(\mathbf{F}_1^1), \mathbf{p}_2^1 = g_2(\mathbf{F}_2^1), \ldots, \mathbf{p}_{K-1}^1 = g_{K-1}(\mathbf{F}_{K-1}^1) \\
&\mathbf{p}_1^2 = g_1(\mathbf{F}_1^2), \mathbf{p}_2^2 = g_2(\mathbf{F}_2^2), \ldots, \mathbf{p}_{K-1}^2 = g_{K-1}(\mathbf{F}_{K-1}^2) \\
&\quad \cdots \\
&\mathbf{p}_1^T = g_1(\mathbf{F}_1^T), \mathbf{p}_2^T = g_2(\mathbf{F}_2^T), \ldots, \mathbf{p}_{K-1}^T = g_{K-1}(\mathbf{F}_{K-1}^T)
\end{aligned}
. \tag{5}
$$

We then apply the cross-entropy loss to all the classification predictions:

$$
\mathcal{L} = \frac{1}{K \cdot T} \left( \sum_{k=1}^{K-1} \sum_{t=1}^{T} \mathcal{L}_{CE}(\mathbf{p}_k^t, y) + \sum_{t=1}^{T} \mathcal{L}_{CE}(\mathbf{p}^t, y) \right). \tag{6}
$$

Table 1: Comparison results with different methods on CIFAR10 and CIFAR100.

| Dataset | Method | Architecture | T | Train Method | Accuracy (%) |
|---|---|---|---|---|---|
| CIFAR10 | SlipReLU (Jiang et al., 2023) | ResNet-18 | 4 | ANN-SNN | 94.59 |
| | Dspike (Li et al., 2021) | ResNet-18 | 6 | SG | 94.25 |
| | TET (Deng et al., 2022) | ResNet-19 | 6 | SG | 94.50 |
| | SLTT (Meng et al., 2023) | ResNet-18 | 6 | SG | 94.59 |
| | PLIF (Fang et al., 2021b) | 7-layer CNN | 8 | SG | 93.50 |
| | TEBN (Duan et al., 2022) | ResNet-19 | 6 | SG | 94.71 |
| | IM-Loss (Guo et al., 2022a) | ResNet-19 | 6 | SG | 95.49 |
| | LSG (Lian et al., 2023) | ResNet-19 | 4 | SG | 95.17 |
| | **DeepTAGE (Ours)** | ResNet-18 | 4 | SG | **95.86** |
| CIFAR100 | SlipReLU (Jiang et al., 2023) | ResNet-18 | 128 | ANN-SNN | 78.55 |
| | Dspike (Li et al., 2021) | ResNet-18 | 4 | SG | 73.35 |
| | TET (Deng et al., 2022) | ResNet-19 | 4 | SG | 74.47 |
| | MPBN (Guo et al., 2023) | VGG16 | 4 | SG | 74.74 |
| | SLTT (Meng et al., 2023) | ResNet-18 | 6 | SG | 74.67 |
| | STBP-tdBN (Zheng et al., 2021) | ResNet-19 | 6 | SG | 71.12 |
| | Sew ResNet (Fang et al., 2021a) | ResNet-34 | 4 | SG | 67.04 |
| | GLIF (Yao et al., 2022) | ResNet-19 | 4 | SG | 77.05 |
| | TEBN (Duan et al., 2022) | ResNet-19 | 4 | SG | 76.13 |
| | **DeepTAGE (Ours)** | ResNet-19 | 4 | SG | **81.39** |

By implementing direct supervision at multiple endpoints within SNNs, our approach further enhances gradient flow on the backpropagation path along multiple network stages and time steps, thereby facilitating the training process and improving performance. Additionally, the auxiliary classification networks introduced can be removed after training without incurring extra computational overhead.

## 4 EXPERIMENTS

In this section, we conduct a thorough analysis and comparison of our approach against other state-of-the-art SNNs using both static and neuromorphic datasets. Furthermore, we validate the effectiveness of the proposed method through extensive ablation studies.

### 4.1 IMPLEMENTATION DETAILS

**Datasets.** We evaluate our models using both static and neuromorphic datasets: CIFAR-10, CIFAR-100 and ImageNet-1k are static datasets, while DVS-CIFAR10 is a neuromorphic dataset. CIFAR-10 consists of 60,000 images of size $32 \times 32$, distributed evenly across 10 classes with 6,000 images per class. CIFAR-100 has 100 classes, with 600 images per class, including 500 training images and 100 testing images. ImageNet-1k is a much larger dataset containing over 1.2 million training images and 50,000 validation images. For static datasets, images are replicated across multiple time steps as input frames. DVS-CIFAR10 is an event-stream dataset derived from CIFAR-10, which captures pixel-level changes in brightness at high temporal resolutions, resulting in a stream of events rather than static frames.

**Training Setup.** We train our SNN model using the Stochastic Gradient Descent (SGD) optimizer with a momentum of 0.9. A cosine learning rate schedule is employed starting from 0.1 and gradually decreasing to 0. The training process involves a batch size of 32 over 320 epochs. Our SNN model employs Leaky Integrate-and-Fire (LIF) neurons, setting the threshold voltage at 1 and the membrane potential decay constant at 2. We utilize an NVIDIA A100 GPU for training and inference on the CIFAR-100 and DVS-CIFAR10 datasets, while four NVIDIA A100 GPUs are used for training on the ImageNet dataset.

Table 2: Comparison results with different methods on ImageNet.

| Dataset | Method | Architecture | T | Train Method | Accuracy (%) |
|---------|--------|--------------|---|--------------|--------------|
| ImageNet | TET (Deng et al., 2022) | ResNet-34 | 6 | SG | 64.79 |
| | MS-ResNet (Fang et al., 2021a) | ResNet-18 | 6 | SG | 63.10 |
| | OTTT (Xiao et al., 2022) | ResNet-34 | 6 | SG | 63.10 |
| | Real Spike (Guo et al., 2022b) | ResNet-18 | 4 | SG | 63.68 |
| | Sew ResNet (Fang et al., 2021a) | ResNet-18 | 4 | SG | 63.18 |
| | MPBN (Guo et al., 2023) | ResNet-18 | 4 | SG | 63.14 |
| | **DeepTAGE (Ours)** | ResNet-18 | 4 | SG | **68.52** |

Table 3: Comparison results with different methods on DVS-CIFAR10.

| Dataset | Method | Architecture | T | Train Method | Accuracy (%) |
|---------|--------|--------------|---|--------------|--------------|
| DVS-CIFAR10 | OTTT (Xiao et al., 2022) | VGG-11 | 10 | SG | 76.30 |
| | SLTT (Meng et al., 2023) | VGG-11 | 10 | SG | 77.30 |
| | STBP-tdBN (Zheng et al., 2021) | ResNet-19 | 10 | SG | 67.80 |
| | MPBN (Guo et al., 2023) | ResNet-19 | 10 | SG | 74.40 |
| | TEBN (Duan et al., 2022) | 7-layer CNN | 10 | SG | 75.10 |
| | LSG (Lian et al., 2023) | ResNet-19 | 10 | SG | 77.90 |
| | **DeepTAGE (Ours)** | VGG-11 | 10 | SG | **81.23** |

## 4.2 COMPARISONS WITH STATE-OF-THE-ART METHODS

**CIFAR10.** Table 1 presents the comparative results of our method with the current state-of-the-art methods on the CIFAR10 datasets. Our method achieves an accuracy of 95.86%, surpassing TET's 94.50% with fewer time steps required. Moreover, our method also demonstrates superior performance compared to the leading ANN-SNN conversion method SlipReLU (Jiang et al., 2023).

**CIFAR100.** Table 1 also details our performance comparison with other methods on the CIFAR100 datasets. Utilizing a spiking ResNet-19 model trained over four time steps, our method attains an accuracy of 81.39%, exceeding the Spatial Learning Through Time (SLTT) (Meng et al., 2023) method by 6.72%. Notably, even the ANN-SNN conversion method SlipReLU (Jiang et al., 2023), which achieves a competitive 78.55% accuracy using 128 time steps, is outperformed by our model in a limited 4 time steps scenario. Additionally, our method surpasses recent advancements like the GLIF (Yao et al., 2022) and TEBN (Duan et al., 2022) methods by significant margins of 4.34% and 5.26%, respectively.

**ImageNet.** Experimental results for the ImageNet dataset using a ResNet-18 backbone are detailed in Table 2. Under direct training, our method surpasses the Sew ResNet (Fang et al., 2021a) by a substantial margin of 5.34%. Furthermore, compared to Real Spike (Guo et al., 2022b) with the same backbone and four time steps, our approach achieves a significant accuracy advantage, reaching 68.52% versus 63.68%. These results demonstrate that our method remains highly effective even in larger-scale datasets and within well-established architectural frameworks.

**DVS-CIFAR10.** Performance evaluations on the DVS-CIFAR10 dataset, utilizing a VGG-11 backbone and spanning ten time steps, are shown in Table 3. Our method achieves an accuracy of 81.23%, significantly outperforming the LSG (Lian et al., 2023) and MPBN (Guo et al., 2023) methods, which report accuracies of 77.90% and 74.40% respectively. These results demonstrate the robustness and efficacy of our approach when applied to neuromorphic data.

## 4.3 ABLATION STUDY

**Temporal-Decoupled Gradient Enhancement and Spatio-Temporal Deep Supervision.** We conduct ablation studies on both static and neuromorphic datasets to evaluate the efficacy of Temporal-Aligned Gradient Enhancement (TAGE) and Spatio-Temporal Deep Supervision (STDS) in our method. In Table 4, we reimplement the Sew-ResNet (Fang et al., 2021a), TET (Deng et al., 2022), and SLTT (Meng et al., 2023) as baselines to evaluate the effect of TAGE across various models. On the CIFAR100 dataset, applying TAGE to the Sew ResNet base model yields an accuracy improve-

Table 4: Ablation studies of TAGE on different methods.

| Dataset | Method | T | Accuracy(%) |
|---|---|---|---|
| CIFAR100 | Sew ResNet (Fang et al., 2021a) | 4 | 77.89 |
| | w/ TAGE | 4 | 78.80 |
| | TET (Deng et al., 2022) | 4 | 79.94 |
| | w/ TAGE | 4 | 80.53 |
| | SLTTMeng et al. (2023) | 4 | 74.46 |
| | w/ TAGE | 4 | 76.36 |
| DVS-CIFAR10 | VGG-11 | 10 | 78.20 |
| | w/ TAGE | 10 | 79.60 |
| | TET (Deng et al., 2022) | 10 | 78.43 |
| | w/ TAGE | 10 | 79.84 |
| | SLTT (Meng et al., 2023) | 10 | 77.67 |
| | w/ TAGE | 10 | 79.75 |

Table 5: Ablation studies of TAGE and STDS in our method.

| Dataset | Model | TAGE | STDS | Accuracy(%) |
|---|---|---|---|---|
| CIFAR100 | Sew ResNet | ✓ | | 78.80 |
| | Sew ResNet | ✓ | ✓ | 81.39 |
| DVS-CIFAR10 | VGG-11 | ✓ | | 79.60 |
| | VGG-11 | ✓ | ✓ | 81.23 |

ment of 0.91 percentage points. Similarly, implementing TAGE on the TET results in an accuracy increase of 0.59 percentage points, while its application on the energy-efficient SLTT improves performance by 1.9 percentage points. These improvements confirm the effectiveness of TAGE in optimizing SNNs across various network architectures and designs. For the neuromorphic DVS-CIFAR10 dataset, the introduction of TAGE into the VGG-11 baseline model raises classification accuracy from 78.20% to 79.60%, demonstrating TAGE's effectiveness for SNN optimization on neuromorphic datasets with larger time steps. In Table 5, we further evaluate the effect of TAGE and STDS in our method, respectively. The accuracy of Sew ResNet increases from 77.89% to 78.80% and then to 81.39%, with the integration of TAGE and then STDS. For the VGG-11 network, the incorporation of TAGE and STDS results in an accuracy increase of 3.03% in total. These results indicate that our TAGE and STDS methods can effectively improve the gradient-based SNN optimization and hence the overall network performance.

**Firing Rate Statistics.** To better evaluate the effect of our Deep Temporal-Aligned Gradient Enhancement (DeepTAGE), we further investigate its impact on the firing rate of SNNs. In Figure 3, utilizing Sew ResNet19 as the backbone architecture, we analyze the firing rates at different layers and time steps both with and without DeepTAGE (denoted as w/ DeepTAGE and w/o DeepTAGE, respectively). As shown in Figure 3(a), DeepTAGE improves firing rates across nearly all network layers, especially in the shallow layers, where improvements are more significant. Moreover, Figure 3(b) demonstrates that firing rates at different time steps have also improved and become more balanced.

**Feature/Class Activation Maps.** In Figure 3(c), we compare the feature activation maps and class activation maps before and after applying our methods (the feature maps of different channels are tiled on a single map for visualization). Compared with the baseline Sew ResNet in the first row, the introduction of TAGE and subsequent DeepTAGE progressively enhance the activations of feature maps, which helps to improve their representational capabilities. Moreover, after using our gradient enhancement methods, the class activation maps are better focused on the key semantic regions of the input image, as shown in the last column of Figure 3(c).

**Training Convergence.** Figures 4 (a) and 4 (b) display the training loss and test accuracy curves over training epochs, comparing scenarios with and without the DeepTAGE method. The training loss decreases more rapidly when applying DeepTAGE, while the test accurate shows a shaper increase, demonsrating that our method significantly speed up training convergence.

**The Hyperparameter $w$.** We also conduct an ablation study on the hyperparameter $w$, which is used to calculate the normalization factor in equation 3. Table 6 presents the test accuracies for

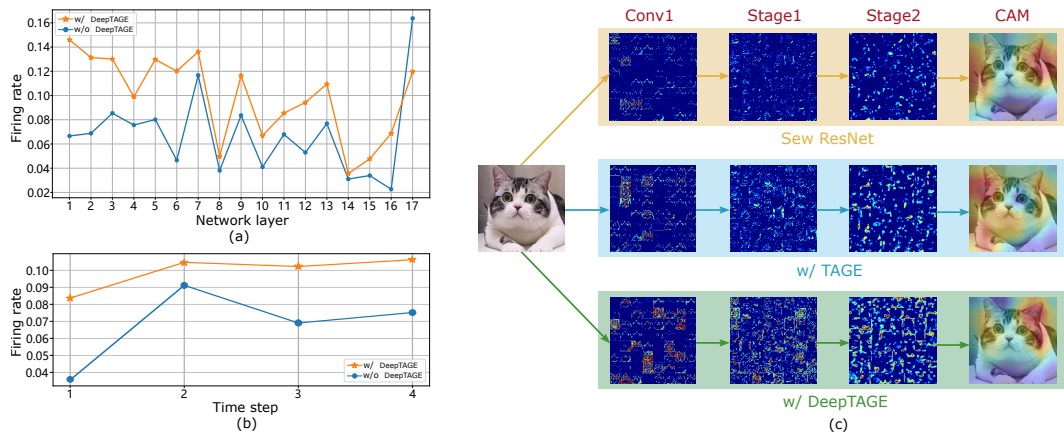

Figure 3: Analysis of firing rates and feature activation maps using Sew ResNet19 as the backbone. (a) Firing rates across different layers with and without DeepTAGE (denoted as w/ DeepTAGE and w/o DeepTAGE). DeepTAGE significantly enhances firing rates, especially in the shallow layers. (b) Comparison of firing rates across time steps, showing more balanced and improved firing rates with DeepTAGE. (c) Visualization of feature activation maps and class activation maps before and after applying TAGE and DeepTAGE.

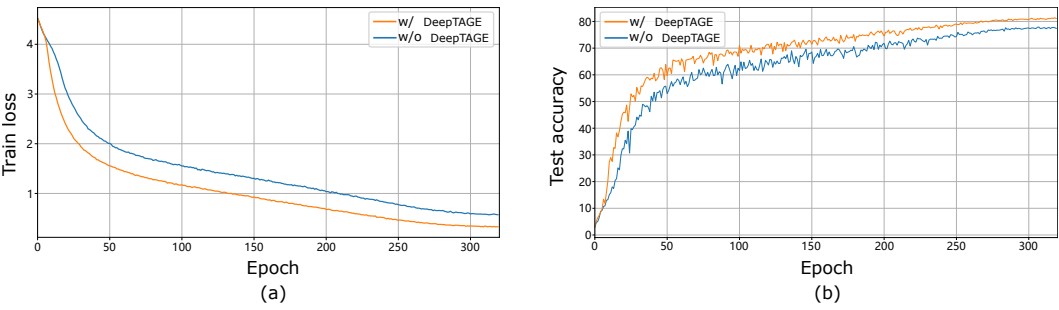

Figure 4: Comparing the loss and accuracy of models with and without DeepTAGE on the CI-FAR100 dataset using the Sew ResNet19 backbone: (a) demonstrates that the model with Deep-TAGE converges significantly faster across epochs compared to the one without. (b) illustrates that the model equipped with DeepTAGE consistently achieves higher performance than its counterpart without DeepTAGE.

Table 6: The ablation studies of $w$ in equation 3.

| Dataset | hyper-parameter | T | Accuracy |
|---|---|---|---|
| CIFAR100 | $w = 1/2$ | 4 | 80.96 |
| | $w = 1/3$ | 4 | 81.02 |
| | $w = 1/4$ | 4 | 81.39 |
| | $w = 1/5$ | 4 | 81.32 |
| | $w = 1/6$ | 4 | 81.17 |
| DVS-CIFAR10 | $w = 1/2$ | 10 | 80.05 |
| | $w = 1/3$ | 10 | 81.14 |
| | $w = 1/4$ | 10 | 81.23 |
| | $w = 1/5$ | 10 | 81.20 |
| | $w = 1/6$ | 10 | 81.21 |

CIFAR100 and DVS-CIFAR10 with different values of $w$. The optimal performance is achieved when $w = 1/4$, which is also the default setting in our method.

## 5   CONCLUSION

In this work, we present the Deep Temporal-Aligned Gradient Enhancement (DeepTAGE) method to tackle critical challenges in optimizing Spiking Neural Networks (SNNs), specifically addressing membrane potential distribution imbalances and vanishing gradients. DeepTAGE dynamically adjusts surrogate gradient functions based on the temporal distribution of membrane potentials, enhancing spike firing efficiency and network convergence. Our approach also incorporates deep supervision at multiple network stages and time steps to strengthen gradient flow, effectively alleviating gradient vanishing issues and boosting overall performance. Extensive experiments on various datasets, including CIFAR10, CIFAR100, ImageNet-1k, and DVS-CIFAR10, show that DeepTAGE outperforms state-of-the-art SNN methods in accuracy without increasing computational costs or adding extra modules. These results confirm the efficacy of DeepTAGE as a scalable and efficient solution for SNN optimization.

## 6   ACKNOWLEDGMENTS

This work was supported by the National Science and Technology Major Project(2020AAA0105801), the National Natural Science Foundation of China (No. 62202469, 62403462,62306312,62036011, 62192782, U2441241), Beijing Natural Science Foundation (L223003), the Project of Beijing Science and Technology Committee (No. Z231100005923046). The Key Research and Development Program of Xinjiang Uyghur Autonomous Region No. 2023B03024.

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
