# OpenReview forum: "DeepTAGE: Deep Temporal-Aligned Gradient Enhancement for Optimizing Spiking Neural Networks"
_ICLR.cc/2025/Conference — ICLR 2025 Poster_

### Official Review · Reviewer_N3KN · 2024-10-31

**Soundness:** 3
**Presentation:** 3
**Contribution:** 3
**Rating:** 8
**Confidence:** 5

**Summary:**

This paper introduces Deep Temporal-Aligned Gradient Enhancement (DeepTAGE), a method for optimizing SNNs. The key idea is to enhance the backpropagation of gradients by dynamically adjusting surrogate gradients based on membrane potential distributions at different time steps. DeepTAGE also incorporates deep supervision across both spatial and temporal levels to ensure balanced optimization. The method is validated through experiments on static (CIFAR10, CIFAR100, ImageNet-1k) and neuromorphic datasets (DVS-CIFAR10), demonstrating performance improvements over state-of-the-art methods without adding computational complexity.

**Strengths:**

1.The paper proposes a well-motivated and novel solution to the improve the optimization of SNNs, backed by comprehensive analysis and empirical validation.
2.The integration of temporal-aligned gradient enhancement and deep supervision is effective, leading to substantial improvements in accuracy without increasing computational costs.

**Weaknesses:**

1.I noticed you tested the gradient enhancement method using the arctangent function. Can this method be generalized to other surrogate gradient functions?
2.Could you provide more details about the structure of the auxiliary classifier networks? Like, how many layers do they have? And how does the number of layers impact the overall network performance?
3.How does deep supervision affect gradient flow in the network? Is there a significant increase in the gradients being backpropagated through the shallow layers?

**Questions:**

Please see weekness

---

### Official Review · Reviewer_j3Kc · 2024-10-31

**Soundness:** 2
**Presentation:** 4
**Contribution:** 3
**Rating:** 6
**Confidence:** 4

**Summary:**

This paper proposes the DeepTAGE method that encompasses two sub-methods. TAGE is used to dynamically adjust the surrogate gradient function based on the membrane potential distribution at different time steps, ensuring that each neuron obtains more effective backpropagation gradients. And STDS alleviates the problem of gradient vanishing by introducing auxiliary network after each SNN stages. On various datasets, DeepTAGE has achieved satisfactory results.

**Strengths:**

Previous related papers often design different surrogate gradients for different layers, ignoring the distribution differences of time steps. But this paper focuses on the adaptability of surrogate gradients at different time steps and provides a good solution. Meanwhile, the results obtained in this paper are very satisfactory, especially the exceptional performance on large-scale datasets. The writing of this paper is clear and easy to understand.

**Weaknesses:**

1.The DTAG method lacks theoretical analysis of how it influence the SNN performance and firing rate.

2.The auxiliary networks introduced by STDS generates extra training overhead. But the authors have neglected the discussion of the extra training cost.

3.Concurrently, there is a lack of analysis regarding how DTAG can ensure the provision of effective gradients to the SNN.

4.DeepTAGE will cause SNN to have an activation frequency, resulting in higher inference energy overhead. How to balance the energy and performance?

**Questions:**

1.How to interpret the sentence on line 168: "Our approach is anticipated to not only stabilize the learning process but also to enhance the representational capacity of SNNs by improving their spike generation capability"?

2.Is DTAG applicable to other types of surrogate gradients?

3.The ImageNet accuracy of ResNet-18, as shown in Table 2 of this paper, is remarkable. DeepTAGE significantly surpasses other methods and is only approximately 1% lower than the reported results of ANN. Could the authors provide the relevant parameters of the model?

4.The accuracies of the spiking ResNet-19 in Table 1 and the Sew ResNet in Table 5 are both 81.39%. Is this a coincidence? What is the base model that Sew ResNet ueses?

---

### Official Review · Reviewer_62ou · 2024-11-01

**Soundness:** 2
**Presentation:** 3
**Contribution:** 2
**Rating:** 6
**Confidence:** 4

**Summary:**

This paper claims increasing latency pushes the distribution of membrane potential away from the threshold, causing a reduction in the performance of the model.  An approach Temporal-Aligned Gradient Enhancement (TAGE), is proposed which adaptively adjusts the surrogate based on the distribution of the membrane potential.

**Strengths:**

The paper is easy to read, writing is good.

The observed phenomenon, shift in the membrane potential distribution away from the threshold with increasing latency is interesting. The proposed method considers this phenomenon and adjusts the surrogate while training.

The authors demonstrate the performance of their approach through empirical results on CIFAR10/100 and ImageNet datasets.

**Weaknesses:**

The phenomenon observed is interesting, however its not clear how and why this distribution shift is affecting the performance of the model. ?

In traditional methods(w/o DeepTage), the firing rate is low compared to DeepTage. Doesn't this cause the increase in energy consumption of the model?

**Questions:**

Could you please provide some results showing does the distribution shift still exists, after the model is trained with the modified surrogate? (Expected it should not exist)

---

### Official Review · Reviewer_Z7sN · 2024-11-02

**Soundness:** 2
**Presentation:** 3
**Contribution:** 3
**Rating:** 5
**Confidence:** 2

**Summary:**

This paper proposes a new approach, Deep Temporal-Aligned Gradient Enhancement (DeepTAGE), which optimizes gradients in SNNs according to the distribution of membrane potentials at each time step. Additionally, it introduces Spatio-Temporal Deep Supervision, incorporating deep supervision across multiple network stages and time steps in SNNs. The effectiveness of DeepTAGE is validated on several popular datasets, achieving SOTA performance.

**Strengths:**

- The paper presents a novel approach to optimizing gradients in SNNs by leveraging both internal surrogate gradient functions and external supervision techniques.
- The experiments are extensive and demonstrate strong results.
- The writing is clear, and the paper is well-organized, making it easy to follow.

**Weaknesses:**

- The motivation for the paper could be strengthened. While the authors present the membrane potential distribution from time steps 1 to 4, it would be beneficial to explore how these distributions evolve over longer time steps. This would emphasize the importance of the problem and make the motivation stronger.
- It would be better if the authors could include the membrane potential distribution after applying TAGE. This comparison would help illustrate the impact of TAGE on stabilizing or shifting the distribution and clarify its effectiveness in optimizing SNN training.
- Regarding the STDS, my intuition suggests that applying CE loss to each stage may actually limit the ability of shallow layers to effectively learn elementary features. This may also create conflicts in the gradient directions across layers. Could the authors provide additional insights into why this approach might enhance training?

**Questions:**

See the 'Weakness'

---

### Meta-Review · Area_Chair_oGCd · 2024-12-22

**Metareview:**

This paper introduces Deep Temporal-Aligned Gradient Enhancement (DeepTAGE) as a new method for optimizing spiking neural networks (SNN). The authors are motivated by solving the vanishing gradient problem brought about by neuronal dynamics and spiking mechanisms in SNNs. By analyzing the dynamic changes of membrane potential distribution, the author proposed to adjust the gradient function according to varying membrane potentials, and combined with the spatiotemporal depth supervision strategy to enhance gradient backpropagation. Experimental results show that DeepTAGE achieves performance improvements on both static and neuromorphic datasets without increasing inference costs. Overall, this article proposes an SNN training enhancement scheme with good experimental results and takes into account the neuromorphic operating overhead of SNN, which can promote the development of SNN learning methods.

After the rebuttal stage, the concerns from reviewers are largely solved. 4 reviewers gave ratings of 5, 6, 6, and 8. Therefore, I recommend this paper to be accepted by ICLR as a spotlight paper.

**Additional Comments On Reviewer Discussion:**

The reviewers all believe that the DeepTAGE method proposed can effectively optimize the training process of SNN, and this method does not need to increase the cost of inference. In addition, the article is easy to understand and performs well on large-scale data sets. However, the reviewers also made some criticisms in their review comments. First, there is insufficient explanation in the paper of how changes in membrane potential distribution affect model performance. In addition, how DeepTAGE balances performance and energy efficiency still requires further analysis. In the rebuttal, the authors explain the reasons for the membrane potential shift and respond to the question of why the method helps improve shallow feature learning. In addition, the author also further discusses the energy consumption issue. I hereby recommend that the author further supplement relevant analysis and experimental data based on the review comments.

---

### Decision · Program_Chairs · 2025-01-22

Accept (Poster)